# Nanotechnology of Tyrosine Kinase Inhibitors in Cancer Therapy: A Perspective

**DOI:** 10.3390/ijms22126538

**Published:** 2021-06-18

**Authors:** Eleonora Russo, Andrea Spallarossa, Bruno Tasso, Carla Villa, Chiara Brullo

**Affiliations:** Section of Medicinal and Cosmetic Chemistry, Department of Pharmacy, University of Genova, Viale Benedetto XV, 3-16132 Genova, Italy; spallarossa@difar.unige.it (A.S.); tasso@difar.unige.it (B.T.); villa@difar.unige.it (C.V.)

**Keywords:** tyrosine kinase inhibitors, nanoparticles, drug delivery, EPR

## Abstract

Nanotechnology is an important application in modern cancer therapy. In comparison with conventional drug formulations, nanoparticles ensure better penetration into the tumor mass by exploiting the enhanced permeability and retention effect, longer blood circulation times by a reduced renal excretion and a decrease in side effects and drug accumulation in healthy tissues. The most significant classes of nanoparticles (i.e., liposomes, inorganic and organic nanoparticles) are here discussed with a particular focus on their use as delivery systems for small molecule tyrosine kinase inhibitors (TKIs). A number of these new compounds (e.g., Imatinib, Dasatinib, Ponatinib) have been approved as first-line therapy in different cancer types but their clinical use is limited by poor solubility and oral bioavailability. Consequently, new nanoparticle systems are necessary to ameliorate formulations and reduce toxicity. In this review, some of the most important TKIs are reported, focusing on ongoing clinical studies, and the recent drug delivery systems for these molecules are investigated.

## 1. Tyrosine Kinase Inhibitors

Intracellular protein tyrosine kinases, including Abelson (Abl), Src, JNK and many others, play a pivotal role in signal transduction pathways and cancer development, being highly activated in malignant tumor cells, but having very low activity and expression in normal cells [1]. Consequently, in the last thirty years, many small molecule tyrosine kinase inhibitors (TKIs) have entered in clinical trials and were approved to treat hematologic and non-hematologic tumors, thus improving cancer treatment. In particular, the greatest progress has been made with the use of TKIs in the treatment of chronic myeloid leukemia (CML). The majority of these molecules are ATP-competitive inhibitors and are not selective, acting also on receptor tyrosine kinases (in particular, platelet-derived growth factor receptor, PDGFR, and vascular endothelial growth receptor, VEGFR) or other intracellular kinases with different selectivity and potency. Unfortunately, all of these new compounds presented sub-optimal properties such as poor solubility (very high pH-dependent solubility), low oral bioavailability and severe adverse effects, which limited their clinical application; in addition, the onset of resistance became the biggest obstacle in clinical application for some of the new molecules (in particular for Imatinib, Figure 1).

Consequently, in recent years many efforts have been made to find new molecules (e.g., Asciminib, Flumatinib; Figure 1) active on resistant CML (in particular on T315I mutation), for the treatment of different diseases (e.g., Ruxolitinib; Figure 1) currently without an effective therapy. In Table 1, selected TKIs (the most important for CML treatment and more recent and innovative than other ones) and their applications are reported. Issues related to solubility and resistance onset could be solved using safe and efficient delivery vehicles, that could improve the therapeutic efficacy, minimize toxicity, ameliorate tumor targetability and decrease drug resistance [2,3].

Imatinib (IM, Gleevec^®^, Glivec^®^) was approved for CML in 2001 and today represents the first-line therapy for this type of hematological tumor, being able to block phosphorylation of Bcr-Abl, a fusion protein kinase which plays a fundamental role in CML development [4]. As IM inhibits PDGFR and c-Kit, two other transmembrane TKs, it has been approved as frontline therapy for: (i) gastrointestinal stromal tumors (GIST), characterized by mutated and over-expressed c-Kit or PDGFR-b [5]; (ii) other myeloid malignancies and hypereosinophilic syndromes and (iii) systemic mastocytosis [6,7]. Today this molecule is the object of 754 clinical trials, 81 of which are in recruitment. Most of them are obviously focused on CML, but also on solid tumors, such as acute lymphoblastic lymphoma (ALL), GIST, melanoma, sarcoma, glioblastoma and papillary thyroid cancer; interestingly, some trials concern asthma (NCT01097694), chronic graft-versus-host disease (GVHD) (NCT01862965), steroid-refractory sclerotic/fibrotic type GVHD (NCT01898377), multiple sclerosis (MS) (NCT03674099) and COVID-19 (NCT04422678), this compound having good immunosuppressive properties. As previously reported, resistance to IM became the biggest problem in its clinical efficacy; in particular, mutations of Bcr-Abl kinase in the ATP-binding site (T315I, M244V, G250E, Y253F/H, E255K/V, M351T and F359V) reduce IM binding affinity versus Bcr-Abl kinase [8] and consequently, second-generation small molecule Bcr-Abl kinase inhibitors (Dasatinib, Nilotinib, Ponatinib, Figure 1) have been developed to overcome this problem [9].

Dasatinib (Sprycel^®^, Figure 1) and Nilotinib (Tasigna^®^, Figure 1), approved in 2006 and 2007 respectively for IM-resistant CML treatment, represent a frontline therapy for patients [10]. Currently, 350 clinical trials (63 in recruitment) regarding Dasatinib are focused on CML, ALL, Hodgkin and non-Hodgkin lymphoma, neck, head, breast, NSCLC, melanoma, mesothelioma, ovarian, colorectal, glioblastoma and CNS tumors (Table 1). In addition, Dasatinib, also acting on PDGFR, Kit, Src, Tek and Btk [11], could be useful as an immunosuppressive agent for immunological disorders [12]. Nilotinib is currently being studied in 219 clinical trials (41 in recruitment, Table 1) to evaluate its efficacy in CML, ALL, GIST and sarcoma (soft tissue sarcoma) patients, but also Huntington’s (NCT03764215), Parkinson’s (NCT02954978) and different forms of dementia pathologies (NCT02947893, NCT04002674).

Sorafenib (Figure 1) is an approved pankinase inhibitor able to target the Ras/Raf/Mek/Erk cascade pathway, PDGFR, VEGFR1/2 and the c-Kit receptor, and to block cell proliferation of different solid tumors; in particular, hepatocellular carcinoma (HCC) [13]. Sorafenib was approved by the FDA as a first-line drug for HCC, liver and kidney cancers. A total of 870 clinical trials (89 in recruitment, Table 1) are focused on HCC, RCC (renal cell carcinoma), liver and kidney cancers, many advanced or metastatic solid malignancies (thyroid, colorectal, neuroblastoma) and different leukemia types. However, its long-term application in clinical practice was hampered by serious dermal toxicity and drug resistance, low water solubility and the first-pass effect [14] and consequent low drug concentration in tumor tissue. Poorly dissolved Sorafenib can also cause large adverse reactions [15,16]. In addition, it can induce paradoxical activation of the MAPK pathway in both malignant and normal stromal cells [17] and this fact in hepatic stellate cells (HSCs) leads to their activation with consequent liver damage. Other pankinase inhibitors (e.g., Sunitinib and Axitinib; Figure 1) have been more recently approved for advanced RCC, unresectable HCC, thyroid cancer and GIST, and many other clinical trials are ongoing also on different solid tumors and leukemia types [18,19].

Seventy-six clinical trials (31 in recruitment, Table 1) focused on CML, ALL and different solid tumors (as NSCLC, GIST, glioblastoma, breast and many others) are reported for Ponatinib, (Iclusig, Figure 1) [20], approved in 2012 for CML treatment. In 2013, the FDA temporarily suspended Ponatinib sales because of the risk of life-threatening blood clots and severe narrowing of blood vessels, but at the end of the same year, this suspension was partially lifted.

Very recently, Novartis announced the results of a phase III ASCEMBL study (multicenter, open-label, randomized study) regarding Asciminib (ABL001, Figure 1), a new Abl allosteric inhibitor; the study evaluated Asciminib administration in adult patients with Philadelphia chromosome-positive CML in chronic phase, previously treated with two or more TKIs for 24 weeks [21,22,23]. On the basis of these interesting results, the FDA has granted Fast Track designation for Asciminib. Now, 14 clinical trials (two of them completed, Table 1) are focused on this compound (alone or in association with IM or Nilotinib) for CML and other leukemic patients; only one clinical trial is focused on asthma treatment (NCT03549897).

Flumatinib (HHGV678, Figure 1) is an orally bioavailable TKI, recently approved in China [24]; it inhibits the wild-type and mutated Bcr-Abl, PDGFR and mast/stem cell growth factor receptor (SCFR and c-Kit). Up to date, five clinical trials (one completed, Table 1) regarding only CML are in progress.

Ruxolitinib (Jakafi, Figure 1) is a selective JAK1 and JAK2 inhibitor approved for myelofibrosis (2011), polycythemia vera (2014) and GVHD in adult and pediatric patients (2019), but now it is also under study for COVID-19 (NCT04414098, NCT04359290, NCT04348071), atopic dermatitis (NCT039208529) and vitiligo (NCT04530344) (Table 1).

Although new compounds are continuously placed on the market and many are effective against different mutations, problems regarding poor solubility, resistance and severe side effects are not completely overcome. In part, the evolutionary probability of resistance can also be overcome with the association of two or more compounds, but this approach does not seem to be conclusive; consequently, the advent of nanotechnologies seems to be of great importance. In addition, it is also possible that the administration of one single nanoparticle containing several drugs may be more effective than the administration of several nanoparticles each containing one compound [25].

## 2. Nanoparticles for Cancer Treatment

The efficacy of conventional chemotherapy based on low-molecular-weight drugs (generally < 1000 Da) is limited by sub-optimal drug pharmacokinetics and biodistribution with off-target accumulation in healthy organs and tissues. The encapsulation of drugs into nanotechnological systems emerged as an effective strategy to overcome these limitations by increasing the renal clearance threshold and blood half-life, reducing kidney excretion and promoting target accumulation [26].

The main advantage of the use of nanoparticles (NPs) is their improved pharmacokinetic (PK) profile compared to small molecule drugs. Small drugs distribute in the total body, resulting in a very large distribution volume (Vd) that requires the use of high doses to achieve the therapeutic drug concentration at the target site, leading to toxic side effects [27]. Nanocarriers have being designed to become entrapped in tumor cells or accumulate around, in order to release the drug safely and specifically to those cells, thus increasing the bioavailability of the drug, while minimizing its exposure to the healthy tissues [28]. The tumor accumulation of nanomedicines is mainly based on the enhanced permeability and retention (EPR) effect [29].

The EPR effect is exploited as a passive target of the carrier since there are openings in the blood vessels and poor lymphatic drainage at the tumor level, which do not occur in healthy tissues. The nanoparticles are highly permeable through tumor vasculatures due to large pores (sometimes reaching several hundred nm to 2 μm) and locally accumulate by dysfunctional and immature drainage in lymphatic vessels, versus normal vasculatures [30]. These features are derived primarily from the rapid growth rate of a tumor and the collapse of existing blood and lymphatic vessels in the occupied space. Passive targeting can be used by nanocarriers because some surface characteristics such as size, shape and zeta potential can be modulated [31]. As reported in a more recent study [32], the EPR effect is more pronounced in murine tumors, whereas in human malignancies, the clinical impact of this effect is less clear and heterogeneous.

Another mechanism of targeted delivery of chemotherapeutic agents was based on the exploitation of tumor cell surface properties, recognized as active targeting. Active targeting is a mechanism that occurs by selective interaction between the nanoparticle transport system and cancer cells. Active-targeting delivery of chemotherapeutic agents is achieved by equipping the nanoparticulate system with suitable ligands, such as antibodies and peptides, to enable their interaction with the tumor cells through molecular recognition [33]. Examples of ligands used for active-targeting nanomedicine formulations to tumor cells are folate [34], transferrin [35] and galactosamine [36]. In this mechanism, targeting ligands are attached on the surface of the nanoparticulate system to facilitate targeting of cancer cells [37]. Functionalization of NPs using a tumor-specific moiety is one of the most widely used techniques to obtain remarkable efficacy and decreased in vivo chemotherapeutic drug toxicity [38].

One of the most frequent problems associated with the use of NPs is the short blood circulation time of an intravenously (i.v.) administered drug delivery system. In this regard, polyethylene glycol (PEG), a water-soluble and biocompatible polymer, is covalently attached to the nanocarrier’s surface in a process known as “PEGylation” [39]. PEGylation enhances the therapeutic drug efficacy by decreasing reticuloendothelial system (RES) uptake. In fact, PEG coating reduces the macrophage uptake by creating a hydration layer that prevents the non-specific adsorption of opsonins onto particles and reduces cell adhesion. Furthermore, PEGylation decreases proteolytic degradation, immunogenicity and kidney clearance and improves solubility, stability and serum half-life [40]. Although PEGylation currently represents the gold standard to reduce NP immunogenicity [41], other materials such as polaxamers, polyvinyl alcohols, polyamino acids and polysaccharides have been used [42]. However, there are ongoing studies investigating alternative approaches to develop long-term circulating NPs with better compatibility and enhanced function than PEGylated NPs. One of them is the biomimetic functionalization of nanoparticles through camouflaging with cellular membranes, that consists of coating the NPs with cell membranes to provide nanoparticles with cell-like behaviors. This approach possesses several advantages, such as prolonged circulation, immune escape and increased targeting abilities [43]. Many types of membranes have been used for constructing cell-membrane-camouflaged NPs, including red blood cells, leukocytes, neutrophils, platelets, macrophages, cytotoxic T cells, stem cells and cancer cells [44]. This type of NPs, coated with a particular cell membrane, will provide homologous targeting and enhanced tumor accumulation.

The uptake of NPs by the cells occurs via phagocytic or non-phagocytic pathways; for example, by endocytosis mechanisms. The controlled release of drugs can occur by: (i) diffusion from the polymeric matrix, (ii) changes in pH and (iii) hydrolysis or enzymatic environment. Another mechanism exploited by stimuli-responsive nanocarriers consists of controlling the drug release profile (as a triggered release) by external factors such as ultrasound, heat, magnetism and light [45,46].

### 2.1. Classification of Nanoparticles

As mentioned above, NPs are an efficient delivery system due to specific biocompatibility, low toxicity and low immunogenicity. In the last years, various types of nanoparticulate delivery systems have been investigated as potential drug carriers. In detail, therapeutic NPs can be divided into two categories: (a) inorganic NPs (e.g., fullerene, quantum dot, metallic nanoparticles); and (b) organic NPs (e.g., polymeric, protein based, micelles, liposomes, solid lipid nanoparticles, dendrimers, nanotubes and nanofibers, nanogels and scaffold matrices) [47].

In this section, the nanoparticulate delivery systems investigated as potential TKI carriers will be described.

#### 2.1.1. Inorganic Nanoparticles

NPs prepared using inorganic materials are widely preferred over organic NPs due to adjustable size and shape manipulation, crystallinity, high surface area, ease of functionalization and high-density surface ligands attachment. Inorganic particles possess peculiar optical, magnetic, catalytic, thermodynamic and electrochemical properties with additional bioactivities. Inorganic particles are classified as metallic and non-metallic NPs. For the anticancer drugs’ delivery, silver, gold and iron oxide metallic particles, and mesoporous silica (PSi) as non-metallic NPs, are the most used (Figure 2).

##### Silica Nanoparticles

Silica nanoparticles are the most common biodegradable inorganic nanomaterials, which can degrade into silicic acid or small silica species under specific aqueous media. Due to their widely accepted biocompatibility, they are considered one of the most promising platforms for biomedical applications, such as drug delivery. In this field, silica nanoparticles have received more attention for the great specific capacity of drug loading. In fact, loaded compounds can be released in a controllable manner with the degradation of silica nanoparticles [49]. Different in vivo results show that the density of silica NPs affected their accumulation in organs, with the high density being correlated with an easy degradation and rapid renal excretion. In addition, by further adsorbing the model drug onto the drug-encapsulated silica nanoparticle, a dual-load drug delivery system was developed [49]. The resulting NPs exhibited a two-phase sustained release with a fast initial step (due to “weak adsorption”) followed by a long-lasting release pattern (ascribable to “relatively strong encapsulation”). Additional advantages, in terms of high thermal stability, chemical inactivity, microbial attack resistance, high hydrophilicity and biocompatibility and high loading capacity make silica an attractive material for biomedical purposes [50]. Despite having many positive aspects, the pulmonary administration of silica NPs may aggravate pulmonary inflammation, asthma and fibrosis [51].

##### Silver Nanoparticles

Silver nanoparticles (AgNPs) are extensively used in biomedical applications for their attractive physiochemical properties, simplicity of preparation in a desired size range with biological functionality and non-toxic nature. In addition, silver-based nanosystems have shown promising antibacterial, antiviral, antifungal and antioxidant properties.

AgNPs allow the coordination of many ligands, enabling easy surface functionalization. They have a special role in modern anticancer therapy, being explored for detection and diagnosis of malignant tumors and as controlled and externally triggered drug delivery systems. As for their antimicrobial activity, their efficacy against cancer cells requires cellular uptake of nanosilver, which can be acquired by diffusion, phagocytosis, pinocytosis and receptor-mediated endocytosis. The size, morphology and surface properties of AgNPs are favorable for internalization by cancer cells, with a consequent local release of silver ions and subsequent oxidative stress induction [52].

##### Gold Nanoparticles

The gold nanoparticles’ (AuNPs) surface functionalization is essential for their functionality, stability and biological compatibility. AuNPs are easy to functionalize with various types of biological molecules or chemical functional groups in order to achieve a suitable vehicle for several purposes, such as diagnostics, cancer therapy and drug delivery systems. Polymeric coating of AuNPs is one of the most common methods of functionalization. As previously reported, PEG is a biopolymer commonly used for surface modification of AuNPs, providing colloidal stability since PEGylated AuNPs repel each other for steric reasons. In detail, PEG is used alone or in association with other biomolecules such as peptides. These functionalized AuNPs can serve as suitable drug vehicles because of their ability to link the cell membrane and penetrating target cells. In addition, AuNPs can be developed as efficient nanocarriers in a drug delivery system (DDS) because of their physicochemical and optical properties, low toxicity and high biocompatibility, as well as their great surface area that allows loading of a high density of drugs. AuNPs can deliver drugs to their targets and control their release by external or internal stimuli [53].

##### Magnetic Nanoparticles

Magnetic nanoparticles (MNPs) of iron oxide, also called magnetite Fe_3_O_4_ or hematite Fe_2_O_3_, have been approved by the US Food and Drug Administration (FDA) for clinical imaging and drug delivery applications. Magnetite is preferred to hematite because of better magnetic properties. These particles are used for imaging purposes in preclinical and clinical studies and their safety and biocompatibility have been established. Easy synthesis, tailoring of shape, size, crystallinity and surface charge, non-toxic nature, magnetic bio-separation and selective targeting along with magnetic fluid hyperthermia revolutionized the use of magnetic nanoparticles for loading a variety of compounds [54]. Easy aggregation is a major problem for MNPs due to their thermodynamic instability and magnetic attraction between particles. Therefore, some organic or inorganic materials were used to coat the MNP surface in order to maintain the stability and reduce their aggregation, causing particle accumulation in deep and inaccessible tissues, and biological interaction at cellular levels [55]. Suitable coating materials on magnetic nanocarriers were reported to play an important role in realizing effective tumor therapy and consequently, magnetic nanoparticles as drug carriers should be used to maximize drug release in tumor areas and minimize drug release in healthy areas [56].

#### 2.1.2. Organic Nanoparticles

The most used organic NPs for drug delivery in cancer therapy are represented by lipid-based nanoparticles (liposomes and lipid NPs, Figure 3), polymer-based nanoparticles and micelles, albumin-based nanocarriers and dendrimers. The main features of each mentioned NP type are presented below.

##### Lipid-Based Nanoparticles (Liposomes and Solid Lipidic Nanoparticles)

Liposomal and lipid-based nanocarriers have been considered as drug carriers for their high drug-loading capacity for both hydrophilic and hydrophobic drugs.

Liposomes, discovered in the 1960s, were the first lipid nanocarriers studied because they are made of the same materials as cell membranes [57]. The liposome is a sphere-shaped hollow vesicle containing an aqueous core and one or more phospholipid bilayers. The first liposome-based drug nanocarriers were Doxil^®^, approved by the FDA for the treatment of Kaposi sarcoma in 1995, and Caelyx^®^, marketed in Europe since 1997. Both formulations consist of a PEGylated liposomal doxorubicin [58]. Differently from niosomes, constituted by non-ionic surfactant and cholesterol [59], liposomes are composed of amphiphilic molecules which are formed of hydrophilic heads and hydrophobic tails. In the presence of water, the hydrophilic portion is arranged outside orienting hydrophobic portion inside. Thus, the resulting bilayer structure is similar to the cell membrane; therefore, it can be well assimilated by cells and consequently gain entrance into cells. Due to their unique advantages, such as good biocompatibility, low biotoxicity, high drug-loading rate and relative stability in vivo, liposomes are always considered as an ideal drug-delivery system. In fact, by modifying different ligands, liposomes can reach the tumor site and release drugs in different ways. Unfortunately, the main disadvantage of liposomes is the rapid clearance from blood and the subsequent premature release of drugs inside the liposome. This is mainly due to the instability of the liposomal structure in blood, the adsorption of proteins and their uptake by macrophages [60]. Liposomes are generally considered to be pharmacologically inactive with minimal toxicity, being composed in most cases of natural phospholipids [61], but in contrast, some studies seem to indicate that liposomes are not immunologically inert as previously suggested [62]. A further advance in the design and development of liposomes is their functionalization to obtain a targeted delivery of the anticancer drug. Examples of targeted liposomes include pH-sensitive systems [63], immunoliposomes functionalized with human monoclonal antibody which binds to neoplastic stomach tissues [64] and the liposomal nanocomplex able to bind to transferrin glycoprotein receptor [65].

Solid lipid nanoparticles (SLNs) were promoted as a safer option compared to other nanosystems, in particular being more stable and cheaper than phospholipid-based liposomes [66]. They can overcome some of the major pitfalls of poor stability and low loading capacity commonly encountered with liposomes, and the possible biotoxicity and residual organic solvent associated with polymeric nanoparticle [67]. SLNs are essentially made of a solid lipid core with a monolayer surfactant shell. Due to its lipidic components, an SLN can solubilize highly lipophilic drugs, and hold them in a stable suspension, avoiding the use of large amounts of surface-active compounds and improving the biopharmaceutical performance after different administration routes.

The advantages of these nanocarriers are related to the feasibility of incorporating both hydrophilic and lipophilic drugs, increased bioavailability of poorly water-soluble molecules, simple sterilization and scale-up procedures. Furthermore, immobilizing drug molecules within solid lipids provides protection from photochemical, oxidative and chemical degradation. The disadvantages include poor drug-loading capacity; in fact, drug encapsulation in SLNs is affected by drug–lipid interaction, nature or state of lipid matrix and drug miscibility with the lipid matrix [68,69]. Lipid-based materials can be utilized with different anticancer drugs, and they are relatively safe and represent a promising and innovative approach for delivery of peptides/protein drugs and genes via different administration routes in tumor therapy.

##### Polymer-Based Nanoparticles

Polymeric NPs can be divided into two types: reservoir systems (nanocapsules), and matrix systems (nanospheres) (Figure 4) [71]. Nanocapsules are composed of a liquid core in which a drug is usually dissolved, surrounded by a polymeric membrane, which controls drug release from the core. Nanospheres are based on a polymeric network in which a drug can be retained inside or adsorbed onto the surface (Figure 4).

Polymeric NPs are prepared from biocompatible polymers and they can deliver drugs in a controlled and targeted way through surface modifications. Polymeric NPs can be synthesized from natural molecules, which include polysaccharide-based (e.g., chitosan, hyaluronic acid, sodium alginate) and protein-based (e.g., gelatin, collagen and albumin) polymers. Furthermore, polymeric NPs can be prepared from synthetic polymers, such as poly acrylic acid (PAA), poly glycolic acid (PGA), poly(lactide-co-glycolide) (PLGA), poly lactic acid (PLA), polyethyleneimine (PEI) and dendrimers. With these synthetic polymers, several types of carriers have been designed, including polymeric micelles, conjugates and NPs.

PLA and PLGA are polyesters largely used in drug delivery due to their biocompatibility and controlled release through the hydrolysis of their ester bonds [72]. In detail, PLA is a polyester homopolymer and PLA-NPs have been used to encapsulate hydrophobic compounds and improve drug solubility [73].

PLGA is a biodegradable polymer used for the development of NPs that can be degraded by the hydrolysis of ester bonds and breakdown into their monomers, lactic acid (LA) and glycolic acid (GA), which are easily excreted from the body [74]. The loading of therapeutic drugs into PLA and PLGA synthetic NPs provides many advantages, such as drug protection from enzymatic degradation and promotion of a sustained release. Recent reports demonstrate that these NPs complexed to anticancer drugs can be used for efficient drug delivery [75] and for reduced nanotoxicity [76].

Micelles are colloidal particles constituted by amphiphilic, lipidic and polymeric micelle molecules [77]. The amphiphilic molecules self-assemble into a structure with a hydrophobic core and a hydrophilic shell [78], forming micelles that typically present diameters of about 100 nm. Micelles go through the fenestrations of tumor vessels and their uptake by the RES system is limited. Their hydrophilic portion defends them from immediate identification by RES and prolongs their circulation time at the systemic level [79,80]. A typical use of these micelles is the encapsulation of hydrophobic drugs that can be introduced into the core of their structure and protected by the hydrophilic shell during delivery to the tumor site.

Finally, dendrimers are hyperbranched molecules that display a low polydispersity and high colloidal stability, a well-defined structure and homogeneous composition [77]. In detail, the core can be composed of polymers such as polypropylene imine (PPI), poly(amidoamine) (PAMAM), polyethylene glycol (PEG) and others. The functional groups of the surface define the anionic, cationic or neutral character of the dendrimer. Some types of dendrimers exhibit inorganic elements as branching points, such as silicon [81] or phosphorus [82], thus identifying carbosilane dendrimers [83] and phosphorhydrazone dendrimers [84] largely used against various types of cancers. In detail, positively charged phosphorus dendrimers have been used as carriers of anticancer drugs (e.g., Cisplatin [85]; Sorafenib [86]). Indeed, the hydrophilic surface and hydrophobic backbone of phosphorus dendrimers make them suitable tools to penetrate membranes.

##### Albumin-Based Nanocarriers

Albumin NPs are protein-based nanocarriers with nonimmunogenic, nontoxic and biocompatible properties that could be used for the delivery of different TKIs [87]. Moreover, their high water solubility and straightforward purification make albumin a versatile drug carrier that can be delivered easily by injection. The use of albumin for clinical purposes dates back to 1940, and in 2005 Abraxane^®^ was approved for metastatic breast cancer [88]. Abraxane^®^ is an albumin nanocarrier of paclitaxel that was studied to maintain the therapeutic efficacy of paclitaxel but minimize the toxicity due to the presence of the surfactant Cremophor^®^ EL in the commercialized paclitaxel formulation (Taxol^®^) [89]. Albumin nanoparticles can be easily prepared, and they are particularly useful for the encapsulation of lipophilic anti-cancer drugs [90].

## 3. Nanoparticles of Tyrosine Kinase Inhibitors

A major part of these new nanoformulations have been patented in the last ten years [91,92]; in general, IM and Dasatinib represent the most studied compounds, whereas new molecules, such as Asciminib, Axitinib and others, are less investigated. Interesting results have been obtained for Sorafenib, Ponatinib and Nilotinib, as reported below. Regarding the routes of administration, these nanocarriers are usually injected intravenously; recent reports describe alternative administration routes thorough intratecal [93] and subcutaneous injection [94].

### 3.1. Imatinib

IM was the first molecule in this series to be nanoformulated and received great interest from researchers. A lot of papers reported the use of IM-encapsulated NPs and demonstrated their efficacy; some of these studies are reported as examples of IM delivery using nanotechnology.

#### 3.1.1. Liposome–Imatinib

One of the first studies on liposome–IM application was focused on the encapsulation of the drug into magnetic liposome nanocomposites, in order to achieve targeted drug delivery in the presence of an alternative magnetic field (AMF) and reduce administration time and dose. The in vitro results demonstrate that AMF strongly promoted IM release from magneto liposome nanocomposites because of nanoparticle motions in the pool of liposome nanocomposite at the applied frequency, owing to an alteration in the permeability of the bilayer. In vivo results suggest magnetically controlled accumulation of liposomes in the targeted sites more rapidly and efficiently [95].

Targeted PEGylated liposomes co-encapsulating IM and paclitaxel were designed to target folate receptors, present on the surface of several types of tumor cells (e.g., ovarian, breast and renal cell carcinomas) [96]; in detail, liposomes were functionalized with folic acid by the extrusion method and lyophilized to standardize their size (mean diameter of 122.85 ± 1.48 nm and polydispersity index of 0.19 ± 0.01). These liposomes had a higher effect on MCF7 cell viability reduction (*p* < 0.05) when compared with non-targeted liposomes and free paclitaxel. In PC-3 cells, viability reduction was greater (*p* < 0.01) when cells were exposed to targeted vesicles co-encapsulating both drugs, compared with the non-targeted formulation.

#### 3.1.2. Lipid Nanocarrier–Imatinib

To overcome some disadvantages presented by liposomes, the researchers studied other lipid carriers that could transport IM effectively to cancer cells.

As an example, Gupta et al. [97] presented the PK aspect and the cytotoxic profiles of NLCs consisting externally of solid lipids (experimental design with 4-factor CCD Central Composite Design considering four different types: Compritol 888 ATO, CmA; Precirol ATO 5, PcA; Gelucire 44/14, GlC; and Sappocire AP, SpA) containing IM dissolved in a liquid lipid (Lubrafil^®^) in the core. The results of this study show that NLCs released the drug in a controlled and prolonged way, since the AUC and t_1/2_ values doubled compared to the classic formulations. Regarding the cytotoxicity results, IM-NLCs exhibited significantly greater cytotoxicity (5 μM, after 48 h incubation) compared to the NLC system without the anti-cancer drug, which was indicative of the high compatibility of the drug delivery system with the cells.

In a recent paper [98], the efficacy of IM was confirmed when encapsulated in lipid nanocapsules (LNCs) prepared by 10% Labrafac and 15% Solutol HS15. In particular, the encapsulation efficiency (99%), particle size (39 nm), zeta potential (−21.5 mV) and release efficiency (60%) of LNCs showed that this system was satisfactory for drug delivery to the tumor level. The in vitro cytotoxicity studies indicated that the drug-free LNCs were nontoxic, whereas the LNCs containing IM showed lower cytotoxicity than the free drug, and the formulation kept the pharmacological activity when the drug was loaded into LNCs. Further in vivo studies will be needed in the future to evaluate the efficacy and side effects of this formulation.

#### 3.1.3. Inorganic Nanoparticles–Imatinib

Among the different types of inorganic nanoparticles, hydroxyapatite, silver and gold NPs will be discussed.

In particular, Sobierajska et al. [99] prepared nanoparticles of hydroxyapatite (nHAp, Ca_10_ (PO_4_)_6_(OH)_2_), whose physicochemical properties were studied by means of XRPD (X-ray Powder Diffraction), SEM–EDS (Scanning Electron Microscopy–Energy Dispersive X-ray Spectroscopy), FT-IR (Fourier-Transform Infrared Spectroscopy) and DLS (Dynamic Light Scattering) techniques. During the preparation of this nanocarrier drug, a conversion from crystalline to amorphous form occurred, contributing to its better solubility and higher dissolution rate in body fluids, improving drug bioavailability. The cytotoxicity of this nanosystem was tested on NI-1, L929 and D17 cancer cell lines; in detail, IC_50_ values for nHAp-IM were similar to those for the drug itself, demonstrating the efficacy of nHAPs

Shandiz et al. [100] prepared IM-loaded silver nanoparticles (IM-AgNPs) by phyto-synthesis starting from a leaves extract obtained from *Eucalyptus procera.* The obtained NPs were characterized by UV–visible spectroscopy, EDS, TEM imaging, SEM, FTIR, DLS (148 nm), zeta potential (−17.4 mV) and drug loading (83–95%) and tested on a human breast cancer cell line. In vitro drug release studies showed a slow and continuous release of IM with an initial burst phase (1–40 h) and a second step (40–80 h) with a constant release profile. This behavior was very important for a more effective therapeutic effect. The viability studies showed that IM-AgNPs significantly diminished the cell viability at different concentrations, which is comparable to the cytotoxicity effect of AgNPs and IM alone. Furthermore, the cytotoxicity studies demonstrated that IM-AgNPs toward the MCF-7 cell line induced cell death by apoptosis rather than necrosis.

The treatment of melanoma using gold nanoparticles coated with alternating layers of chitosan (CS) and sodium alginate (SA) containing anti STAT3, siRNA and IM mesylate was presented [101]. These nanoparticles were characterized by size (150–170 nm) and zeta potential (−33/−77 mV), encapsulation efficiency (about 60%) and release profile; the IM release from nanoparticles was controlled for up to 36 h vs. 100% free IM within 2h. An additional in vivo study was also carried out on six mice to which the formulations were administered topically (passive or iontophoresis application) or by intratumoral injection. The results obtained show that the codelivery of STAT3 siRNA and IM using AuNPs showed a significant (*p* < 0.05) reduction in percentage tumor volume, tumor weight and suppressed STAT3 protein expression. In addition, topical iontophoretic administration of the nanoparticle system seems to be comparable to intratumoral administration.

#### 3.1.4. Polymeric Organic Nanoparticles–Imatinib

Poly(butyl)cyanoacrylate (PBCA) nanoparticles stabilized with PEG and containing IM mesylate have been developed and biologically evaluated on the leukemia cell line K562 using MTT assay [102]. Size (224 nm), zeta potential (−9.5 mV), encapsulation efficiency (86%), sustained drug release (10% in 48 h), cytotoxicity and stability (2 months) were investigated and confirmed the suitability of PBCA nanoparticles for IM delivery in the K562 leukemia cell line, with IC_50_ values for the encapsulated drug and free drug of 6 and 10 µM, respectively.

CS nanoparticles were prepared by ionic gelation technique and using an experimental design [103]. An optimized formulation, showing a particle size of 208 nm, zeta potential of −32.56 mV, in vitro cumulative drug release of 86% after 80h and drug entrapment efficacy 68%, has been identified. This optimized preparation was engineered to target colorectal cancer cell lines. Using MTT assay, the authors confirmed that IM-NPs had a significant, sustained in vitro anticancer activity against the CT26 cell line and showed no sign of damage in tissues, indicating that the final formulation can be safely administered through an i.v. route.

### 3.2. Dasatinib

Many patents focus on Dasatinib nanoformulations and most of them are innovative and very recent [104,105,106]. As previously reported, a major problem for oral administration of Dasatinib is its poor bioavailability caused by a low solubility, inappropriate partition coefficient, low drug permeation through lipid membrane, first-pass metabolism, P-glycoprotein-mediated efflux and drug degradation in the gastrointestinal tract due to the pH of the stomach or enzymatic degradation [107]. Animal data suggest that, due to an extensive first-pass effect, the bioavailability of Dasatinib is about 14–34%. Thus, limited aqueous solubility is the bottleneck for the therapeutic outcome of Dasatinib. The majority of Dasatinib nanoformulations have been developed to treat CML cell lines, but as reported below, also for solid tumor treatment. In addition, in the last years Dasatinib-loaded NPs have been developed for different diseases, in particular ocular diseases (proliferative vitreoretinopathy, PVR) [108,109].

#### 3.2.1. Solid Lipid Nanoparticles (SLNs)–Dasatinib

To overcome hepatic first-pass metabolism and to enhance oral bioavailability, lipid-based DDSs such as SLNs can be used. Mohamed et al. [110] prepared Dasatinib-loaded NPs to improve the oral bioavailability by exploiting the intestinal lymphatic transport. The obtained results show that bioavailability of Dasatinib–SLN was 2.28-fold higher than a simple Dasatinib suspension. The same authors investigated enhanced solubility and bioavailability of Dasatinib by incorporating it with different lipids, such as Dynasan 114, 116 and 118, and exploited its intestinal lymphatic transport. These Dasatinib-loaded SLNs were prepared by hot homogenization followed by the ultra-sonication method and were characterized for particle size, polydispersity index, zeta potential, encapsulation efficiency, total drug content and in vitro release [111].

Begum et al. [112] prepared different formulations, using many surfactants to avoid aggregation and to stabilize SLNs. Dasatinib-loaded SLNs have been evaluated for study of drug excipients compatibility, polydispersity index, particle size, surface morphological, zeta potential and drug release features. Stability studies revealed that these SLN dispersions stored in 4 °C were stable.

#### 3.2.2. Inorganic NPs–Dasatinib

Adena et al. [113] prepared Dasatinib-loaded AuNPs and performed risk assessment, optimization, in vitro characterizations, stability and drug release studies and cytotoxicity and in vivo pharmacokinetic evaluation. The prepared AuNPs showed enhanced growth inhibition percentages and systemic bioavailability in comparison with Dasatinib alone, and therefore represent a promising delivery system in the treatment of CML. The same authors also reported a new process for the preparation of AuNPs using chitosan and PEG coating of the particle surface, and entrapping Dasatinib in the chitosan–PEG surface corona. In this way, they obtained Dasatinib-loaded PEG-stabilized CS-capped AuNPs stable at different storage conditions and endowed with sustained drug release of up to 76% in 48 h [114].

Li et al. [115] developed an interesting and efficient method for the production of linear Dasatinib−DNA conjugates in good yields via “click chemistry”, which upon further conjugation with AuNPs, generate therapeutically important bi-functional particles. Dasatinib−DNA conjugates, as well as the derived AuNPs, show equal efficacy toward leukemia cells. These two approaches may be extended to the synthesis of other drug−nucleic acid conjugates for further biological studies.

Other investigations described Dasatinib-loaded AuNPs selectively activated by the presence of specific mRNA. In detail, Dasatinib was conjugated with a specific oligonucleotide complementary (anti-sense) to an mRNA overexpressed in cancer cells (human BIRC5 mRNA, highly expressed in many cancers, and AML1/ETO translocation, present with high frequency in acute myeloid leukemia) [116,117]. The authors demonstrated both the efficient delivery and the selective release of Dasatinib from AuNPs in leukemia cells with good in vitro and in vivo efficacy. Furthermore, these AuNPs reduce toxicity against hematopoietic stem cells and T cells. Targeted NPs have the potential to improve drug therapeutic efficacy and minimize toxicity, being highly customizable on the cancer cell mRNAs [118].

Other inorganic particles are mesoporous silica NPs, which were recently investigated to enhance the poor solubility of Dasatinib. Vadia et al. [119] reported a Dasatinib-loaded mesoporous silica nanoparticulate system with a remarkable dissolution enhancement, when compared to the pure drug.

Sally et al. [120] very recently reported Dasatinib-loaded magnetic micelles (MNPs) composed of a hydrophobic oleic-acid-coated magnetite (Fe_3_O_4_) core along with Dasatinib encapsulated in amphiphilic zein-lactoferrin self-assembled polymeric micelles. These NPs showed good in vitro serum stability and hemocompatibility with a sustained release of Dasatinib in acidic pH and showed a 1.35-fold increased cytotoxicity against the triple-negative human breast cancer cell line (MDA-MB-231). Overall, these results suggest that Dasatinib-loaded MNPs possess a great potential for breast cancer targeted therapy.

#### 3.2.3. Polymeric Organic Nanoparticles–Dasatinib

Regarding organic polymeric NPs, Valero et al. [121] synthesized 500 nm cross-linked polystyrene nanospheres to obtain bifunctionalized doubly PEGylated NPs, labeled in turn with a far-red fluorescent dye (named Cy5). The obtained NPs were conjugated with a Dasatinib analogue (obtained by “click chemistry” reaction and endowed with similar biological activity to Dasatinib) to give new Dasatinib-decorated Cy5-tagged nanospheres. These particular NPs were efficiently internalized into living MDA-MB-231 cells and induced human recombinant Src activity in vitro and in cell inhibition, proving the capacity of this novel nanodevice to detect drug–target interactions in complex living environments. Other authors reported new polymeric Dasatinib-loaded nanocarriers with a superior efficacy in several cell lines representative of triple negative breast cancer (MDA-MB-231, HS578T and BT549) in comparison with the free drug. In detail, these NPs were formulated using a biodegradable and biocompatible polyester. In this paper, enzymatic and cellular degradation of the new drug delivery system were studied, and the toxicity and blood compatibility were evaluated for a potential clinical use [122]. An interesting study reported an analysis concerning the use of excipients able to synergize the action of active pharmaceutical ingredients. Although a large spectrum of surfactants can be used for the preparation of polymeric NPs, the poloxamer surfactants were shown to preferentially target cancer cells, as well as inhibit multidrug-resistant proteins and other drug efflux transporters on the surface of cancer cells. The aim of this study was to assess the effect of three different poloxamer surfactants (i.e., Pluronic F-108, Pluronic F-127 and Kolliphor P-188) and chitosan on the stability, immunogenicity and cytotoxicity of Dasatinib-loaded NPs, as well as cellular uptake of NPs by various cell types. Poloxamers were observed to be non-toxic for the cells; however, stimulation of cell growth was evidenced in some cases. The properties exhibited by the various poloxamers in this study could guide the selection of an appropriate poloxamer NP formulation [123]. In another study, a simple PEGylated peptidic nanocarrier was developed for effective co-delivery of Doxorubicin and Dasatinib combination chemotherapy. Co-encapsulation of the two agents was facilitated by incorporation of 9-fluorenylmethoxycarbonyl (Fmoc) and carboxybenzyl (Cbz) groups into a nanocarrier for effective carrier–drug interactions. Synthesized micelles were more effective than other treatments in inhibiting the proliferation and migration of cancer cells; particularly, a tumor growth inhibition rate of 95% was achieved at a respective dose of 5 mg/Kg for Doxorubicin and Dasatinib in a murine breast cancer model [124].

Multilayered, multifunctional polymer coatings grafted onto carbon nanotubes (CNTs) were reported by Moore et al. [109] to control the kinetics of Dasatinib release and its therapeutic efficacy. In detail, biocompatible, biodegradable multilayered coatings composed of PGA and PLA were polymerized directly onto hydroxyl-functionalized CNT surfaces. Sequential addition of monomers into the reaction vessel enabled multilayered coatings of PLA–PGA or PGA–PLA. PEG capped the polymer chain ends, resulting in a multifunctional amphiphilic coating. These multilayer polymer coatings on CNTs controlled Dasatinib release and enhanced in vitro therapeutic efficacy against the U-87 glioblastoma cell line, compared to monolayer polymer coatings [125].

#### 3.2.4. Targeted NPs–Dasatinib

As for others TKIs, also for Dasatinib many targeted NPs have been reported. Recent structural activity reports suggest that Dasatinib can be modified on its hydroxy terminus, without affecting its therapeutic potential, and this fact has been exploited by different authors to obtain functionalized and targeted NPs [126].

As previously reported, albumin NPs may serve as useful drug carrier for cell-selective drug delivery to reduce drug-induced endothelial hyperpermeability and to maintain endothelial barrier integrity. In an interesting study, authors reported Dasatinib-loaded NPs with increased anti-leukemia efficacy in comparison with free Dasatinib. Importantly, albumin NP as a drug carrier markedly reduced Dasatinib-induced endothelial hyperpermeability by restraining the inhibition of the Lyn kinase signaling pathway in endothelial cells. Therefore, albumin NPs could be a potential tool to improve Dasatinib anti-leukemia efficacy through its cell-selective action [127].

Kim et al. [128] reported a novel technology, called “InCell IT”, for in situ monitoring of bindings between a small molecule kinase inhibitor (e.g., Dasatinib) and its target protein kinases in live cells. In detail, Streptavidin-attached MNPs were coated by biotinylated Dasatinib; these Dasatinib-magnetic NPs were transferred into cells (HeLa cells, expressing the target proteins fused with a green fluorescence protein used as a tag for detection). Through this technology, the authors demonstrated the binding between Dasatinib and its target protein kinases (including Src, ABL1 and CSK) [128].

More recently, many authors reported NPs functionalized with Matrix Metalloproteinase 2 (MMP2), an extracellular enzyme upregulated and produced by cancer cells and various tumor stromal cells in the tumor microenvironment, involved in cancer initiation, growth and metastasis; in addition, MMP2 binds to cancer and endothelial cells to facilitate cancer invasion and angiogenesis [129]. Yao et al. [130,131] synthesized a series of analogues of polyethylene glycol-MMP2-cleavable peptide–phosphatidylethanolamine (PEG-pp-PE) copolymers and revealed the relationship between their chemical structure and the activity against P-gp-induced drug efflux in multidrug-resistant cancer cells (MDR) [130,131]. In a subsequent study, authors developed an MMP2-sensitive polymer (PEG2k-pp-PE), to construct Dasatinib micellar NPs with three specific objectives: drug delivery, tumor targeting and sensitization of resistant cancer cells to drug treatments. The results indicate that the PEG2k-pp-PE polymers and micelles could inhibit the drug efflux, facilitate cellular uptake and penetration and increase drugs’ tumor targeting and retention, leading to improved anticancer activity. This study demonstrated that PEG2k-pp-PE micelles might have great potential to be a multifunctional nanocarrier for effective cancer treatment [132].

More recently, the same authors prepared and characterized similar types of Dasatinib-loaded NPs, with tumor-associated macrophages (TAM) selectivity. In detail, macrophage selectivity, tissue penetration and macrophage-killing capability of prepared NPs were evaluated in the individual cell cultures, cancer cell/macrophage cocultures, three-dimensional (3D) coculture spheroids, and zebrafish. Finally, the biodistribution, TAM-targeting anticancer activity and side effects were evaluated in in vivo tumor-bearing mice. Interestingly, the NPs developed in this study demonstrated a great accuracy and efficiency in TAM targeting and drug delivery [133].

Tosi et al. [134] synthesized a series of [^18^F] Dasatinib derivatives and [^89^Zr]-labeled peptide nanofiber Dasatinib-conjugates and evaluated their efficacy against glioma and their ability to be retained in brain tissue by positron emission tomography (PET). This PET study allowed a quantitative determination of the distribution of Dasatinib in the brain tissue and defined the relationship between specific molecular alterations and drug clearance.

In 2020, ultrasmall fluorescent core-shell silica NPs (named Cornell prime dots, C′ dots), functionalized with various densities of cyclized Arg–Gly–Asp (cRGD) peptides, were developed to target αv integrins, a well-described target of the tumor neovasculature and gliomas [135]. The utility of these NPs for enhancing accumulation, distribution and retention (ADR) in a genetically engineered mouse model of glioblastoma was evaluated. By linking these NPs to Dasatinib, authors produced new Dasatinib-loaded NPs with improved brain tumor delivery and penetration, enhanced ADR in glioma cells and efficient drug delivery, as demonstrated by the inhibition of different downstream pathways. These results evidenced that highly engineered C′ dots are promising drug delivery vehicles able to overcome the complex physiological barriers typical of the brain [136].

### 3.3. Nilotinib

Nilotinib (Tasigna^®^) is a recent pankinase inhibitor used for IM-resistant CML. In addition, a number of patents have been published on nanoformulations prepared to ameliorate Nilotinib activity and efficacy [137,138].

Very recently, Koehl et al. [139] explored the potential of lipid vehicles to improve the bioavailability of hydrophobic drugs such as Nilotinib, comparing a chase-dosing approach and lipid suspensions. To improve the dissolution kinetics, gastrointestinal absorption and bioavailability of some TKIs (including Nilotinib), Jesson et al. [140] prepared hybrid NPs, consisting of amorphous TKI embedded in a polymer matrix; these nanosystems displayed an increase in Nilotinib release rate in both simulated gastric fluid and intestinal fluid, particularly when surfactants are present on the hybrid nanoparticle surface. The prepared hybrid NPs represent a promising approach to improve drug dissolution rate, gastrointestinal absorption and bioavailability following oral administration [140].

Other targeted nanoformulations have been developed to minimize the resistance phenomenon and reduce cytotoxicity, not only for CML treatment, but also for application in different solid tumors. Recently, Fan et al. [141] prepared collagenase I and retinol co-decorated polymeric micelles that possess a nanodrill-like and HSCs-targeting function based on poly(lactic-co-glycolic)-poly(ethyleneglycol)-maleimide (PLGA-PEG-Mal) (named CRM) for liver fibrosis treatment. These particular functionalized NPs could realize excellent accumulation in fibrotic liver and accurate targeting to activated HSCs in a mouse hepatic fibrosis model. Moreover, CRM loaded with Nilotinib showed optimal antifibrotic activity, suggesting that CRM is an efficient carrier for liver fibrosis drug delivery; in this study, the authors demonstrated that collagenase I, decorating NPs, could be a new strategy for building a more efficient HSCs-targeting nanodrug delivery system [141].

In another work, wool-like NPs were developed to treat CML; in detail, a poly(ε-caprolactone) (PCL) nanosystem, composed of a biodegradable pH-sensitive core releasing Nilotinib and an enzyme-sensitive outer shell releasing IM mesylate, were prepared. This combinatorial delivery showed reduced IC_50_ values on leukemia cells compared to single free drugs administration. In addition, in vitro results evidence a consistent drug release and a more therapeutic efficiency at a low dose with respect to the single-drug nanoformulation, confirming that both drugs reached the target cell precisely, maximizing the cytotoxicity and minimizing drug cell resistance [142].

### 3.4. Ponatinib

As reported for Nilotinib, many patents were focused on nanoformulation of Ponatinib [137,138,143] and different authors published various methods to obtain Ponatinib formulations [144].

Targeted Ponatinib-loaded NPs have been recently investigated to evaluate their biological effect on osteosarcoma cell lines. In detail, Zinger et al. [145] reported the design and synthesis of biomimetic/targeted NPs incorporating Ponatinib. These SLNs incorporate membrane proteins purified from activated leukocytes that enable immune evasion and enhanced targeting of inflamed endothelium. The NP formulations showed promising dose−response results in two different murine osteosarcoma cell lines, indicating efficient Ponatinib loading and a possible application for numerous therapeutic agents with toxicity profiles.

Kallus et al. [146] encapsulated Ponatinib and Nintedanib into liposomes to obtain increased tumor accumulation/specificity and reduced side effects. Different methods of drug loading were tested and, interestingly, in an FGFR inhibitor-sensitive murine osteosarcoma transplantation model (K7M2), only liposomal, but not free, Ponatinib showed a significant tumor growth inhibition with reduced side effects.

### 3.5. Sunitinib

Regarding Sunitinib, a major part of efforts has been focused on the production of polymeric nanoparticles. In addition, a more recent Chinese patent is focused on PLA–PEG–PLA Sunitinib NPs [147].

Otroi et al. [148] prepared Sunitinib-loaded poly (3-hydroxybutyrate-co-3-hydroxyvalerate acid) NPs, obtaining dry powders after spray drying, and evaluated their cytotoxicity effects on A549 cells by MTT assay. This formulated inhalable powder could represent a promising medication for local therapy of lung cancer.

Nanopolymeric pharmaceutical excipients, such as CS nanoparticles, were synthesized and evaluated as in vitro drug release systems for Sunitinib [149].

Shi et al. [150] developed different targeted liposome formulations able to treat resistant breast cancer in vitro. In detail, targeted Sunitinib plus vinorelbine liposomes showed a good inhibitory effect on resistant MCF-7/Adr cells and represented a novel type of nanoformulations, which could accumulate in the resistant breast cancer cells.

In another study, the same authors developed a novel type of targeted liposomes by modifying a mitochondriotropic material (i.e., D-a-tocopheryl polyethylene glycol 1000 succinate–triphenylphosphine conjugate, TPGS1000-TPP), to encapsulate Sunitinib. Biological in vitro evaluations were carried out, in breast cancer cell lines (MCF-7 and MDA-MB-435S) and in vivo in mice. Targeted drug liposomes were internalized via cellular uptake and accumulated in the mitochondria of invasive breast cancer cells, inducing acute cytotoxic injury and apoptosis [151]. Interestingly, other studies showed successful development of Sunitinib-loaded PLGA-NPs not only for cancer therapy, but also for neovascular age-related macular degeneration disease [152].

### 3.6. Sorafenib

To overcome the delivery problems previously presented for Sorafenib, many efforts have been made to obtain different NPs, particularly SLN [153], graphene nanosheets [154], PSi and AuNPs in a polymeric nanocomplex [155], PLA NPs [156], hydroxypropylmethylcellulose (HPMC) or polyvinyl pyrrolidone and poloxamer NPs [157], dextran and poly(-lactide-co-glycolide) [DexPLGA] NPs [158]. Overall, these studies showed the importance of systematic formulation design to overcome poor solubility of the drug. In comparison with free Sorafenib, the majority of the reported Sorafenib NP formulations exhibited a significant increase in the retention time, a higher drug concentration in tumor tissues and an increased efficacy in inhibiting tumor growth.

Even more interesting results have been obtained with design, synthesis and biological evaluation of targeted Sorafenib nanoformulations. Wang et al. [159] prepared PSi nanoparticles functionalized with a specific peptide able to direct Sorafenib to the tumor tissue and thus enhance the cellular uptake and drug delivery efficiency. In detail, the targeting peptides were obtained by azide alkyne cycloaddition click reaction, an important tool for surface modification of nanomaterials. The new Sorafenib-loaded targeted NPs efficiently delivered the drug into the cells, resulting in enhanced in vitro antiproliferative activity, and should represent an interesting system for targeted cancer therapy.

Hong et al. [160] developed CXCR4-targeted NPs specific for activated HSCs in fibrotic livers. CXCR4 is a chemokine receptor induced in HSCs by various cellular stresses during the progression of liver fibrosis. As Sorafenib treatment could attenuate liver fibrosis and was associated with the inhibition of angiogenesis, the authors examined the anti-angiogenic activity of Sorafenib and Selumetinib (a MEK inhibitor) co-formulated in peptide-modified NPs (constituting PLGA, dipalmitoyl phosphatidylcholine, peptides (CTCE9908) and PEG). In mice with CCl_4_-induced liver fibrosis, treatment with Sorafenib/Selumetinib-loaded CXCR4-targeted NPs significantly suppressed hepatic fibrosis progression and further prevented fibrosis and liver metastasis [161,162].

Other polymeric or targeted NPs have been recently reported for the treatment of HCC. As an example, Yu et al. [163] designed and synthesized bovine serum albumin (BSA)-coated zinc phthalocyanine (ZnPc) and Sorafenib NPs (ZnPc/SFB/BSA) able to trigger photodynamic therapy (PDT), photothermal therapy and chemotherapy. Upon irradiation at 730 nm, these NPs significantly suppressed HCC cell proliferation and metastasis and promoted in vitro cell apoptosis, with low toxicity and adequate blood compatibility. In addition, injection of ZnPc/SFB/BSA reduced tumor growth in a xenograft HCC model. All these results confirm that this type of nanoformulation could represent a promising strategy for HCC patients.

Recently, Li et al. [164] developed new Sorafenib-loaded dendritic polymeric NPs with excellent stability, high cellular uptake efficiency in HepG2 human liver cells and higher cytotoxicity than free Sorafenib. Furthermore, this NP formulation inhibited tumor growth in mice bearing HepG2 xenografts, with negligible side effects, thus representing a novel approach for enhanced therapy of HCC.

Sorafenib-loaded polymeric NPs (constituted by TPGS-b-PCL copolymer) were synthesized from ε-caprolactone and D-α-tocopheryl polyethylene glycol 1000 succinate via ring-opening polymerization. The obtained NPs contained Pluronic P123 modified with anti-GPC3 antibody (NP-SFB-Ab) and displayed good stability, high drug release into cell culture medium and improved cytotoxicity in comparison with non-targeted NP-Sorafenib and the free drug. In addition, these targeted NPs significantly inhibited the growth of HepG2 xenograft tumors in nude mice without producing side effects. These findings suggest NP-SFB-Ab as a promising new method for achieving targeted therapy in HCC [165].

The association of Sorafenib with other different therapeutic agents has also been investigated for HCC treatment. Cao et al. [166] reported the co-delivery of Sorafenib and curcumin by directed self-assembled NPs. This nanosystem was prepared taking advantage of the hydrophobic interactions among Sorafenib, curcumin and the hydrophobic segments of PEG derivatives of vitamin E succinate. This innovative nanoformulation showed in vitro cytotoxicity and cell apoptosis in BEL-7402 and Hep G2 cells, in addition to a good antiangiogenetic action.

Zhang et al. [167] reported Paclitaxel- and Sorafenib-loaded albumin nanoparticles to avoid taxol toxicities and to evaluate the anticancer efficacy of this combination. Interestingly, the authors obtained lower myelosuppression and hemolysis and an increased antitumor effect in animal models.

The co-delivery of plantamajoside (natural herbal medicines with excellent antiproliferative effect against many drug-resistant cancers) and Sorafenib by multi-functional PLA NPs was investigated to overcome drug resistance in HCC. NPs were produced and co-loaded with Sorafenib and plantamajoside and decorated with a polypeptide which specifically binds to biomolecules overexpressed on the surface of cancer cells. The authors demonstrated that this functionalization improved drug accumulation and penetration at tumor sites, resulting in a strong inhibition of tumor growth [168]. Li et al. [169] formulated a dual-targeting delivery system for enhanced HCC therapy by encapsulating Sorafenib and anti-miRNA21 (an antisense oligonucleotide with great potential in cancer therapy) in pentapeptide-modified reconstituted high-density lipoprotein NPs. In addition, these NPs expressed apolipoprotein A-I (ApoA-I) which specifically binds to overexpressed scavenger type B1 receptor (SR-B1) present in HCC parenchyma. This nanosystem was able to drive loaded drugs simultaneously to tumor neovascular and parenchyma, achieving precise delivery of therapeutics to maximize the efficacy. At the targeted sites, anti-miRNA21 would assist Sorafenib to exert powerful anticancer and anti-angiogenetic effects. The obtained results evidence that this chemo-gene system significantly increased Sorafenib action with negligible toxicity and reversed drug resistance, with improved efficacy in HCC [170].

Other nanoformulations have been developed to target different solid tumors, such as renal cell carcinoma (RCC). Liu et al. [171] prepared different Sorafenib-loaded PLGA, 1,2-dipalmitoyl-sn-glycero-3-phosphocholine (DPPC) liposomes, and hydrophobically modified chitosan-coated DPPC liposomes with good action against RCC 786-0 renal cancer cells. Poojari et al. [172] assembled layer-by-layer (LbL) polyelectrolytes dextran-sulfate/poly-L-arginine with Sorafenib-encapsulated calcium carbonate NPs for oral cancer therapy. This innovative nanoformulation exhibited more potent antiproliferative, apoptotic and antimigratory activities in KB cells than the free drug, providing new insights for oral cancer therapy.

## 4. Conclusions

In the last 20 years, the discovery of small molecule TKIs has led to a significant improvement in anti-cancer therapy, especially for the treatment of CML, but also for other solid tumors. Unfortunately, the onset of resistance, the associated side effects and the poor bioavailability have limited the use of this class of drugs. To overcome these limitations, several drug delivery systems have been developed. From the data available to date, it emerged that NPs led to a significant improvement in the effectiveness of TKIs, with an amelioration of bioavailability and a decrease in side effects, obtaining more targeted therapies. Therefore, NPs represent a new starting point for the development of targeted, more powerful and less invasive anti-cancer therapies, which can also re-evaluate compounds with drug delivery problems.

## Figures and Tables

**Figure 1 ijms-22-06538-f001:**
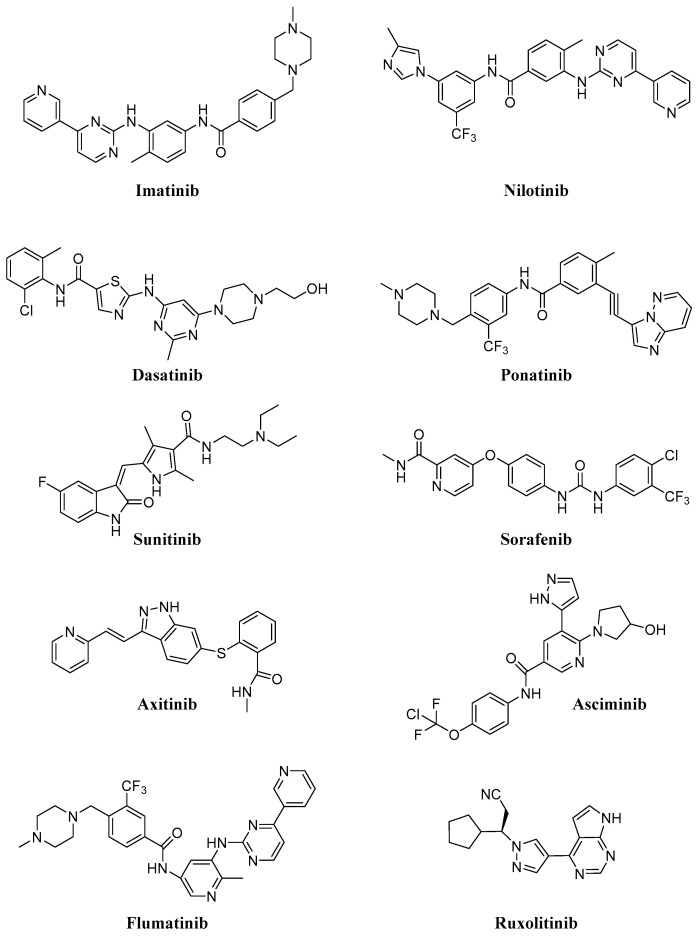
Chemical structure of selected tyrosine kinase inhibitors (TKIs).

**Figure 2 ijms-22-06538-f002:**
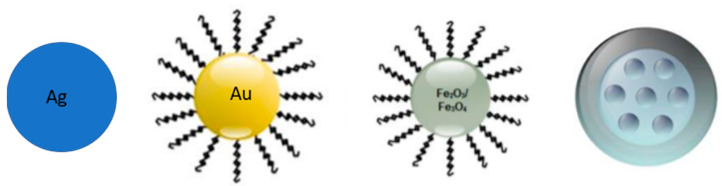
Schematic representation of inorganic nanoparticles (AgNPs, AuNPs, MNPs and PSi NPs) [48].

**Figure 3 ijms-22-06538-f003:**
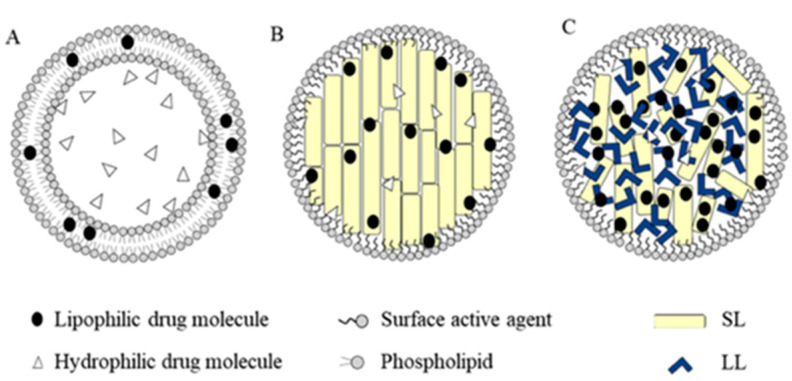
Different types of lipid-based nanoparticles. (**A**) Liposomes; (**B**) solid lipid nanoparticles (SLNs); (**C**) nanostructured lipid carriers (NLCs) [70].

**Figure 4 ijms-22-06538-f004:**
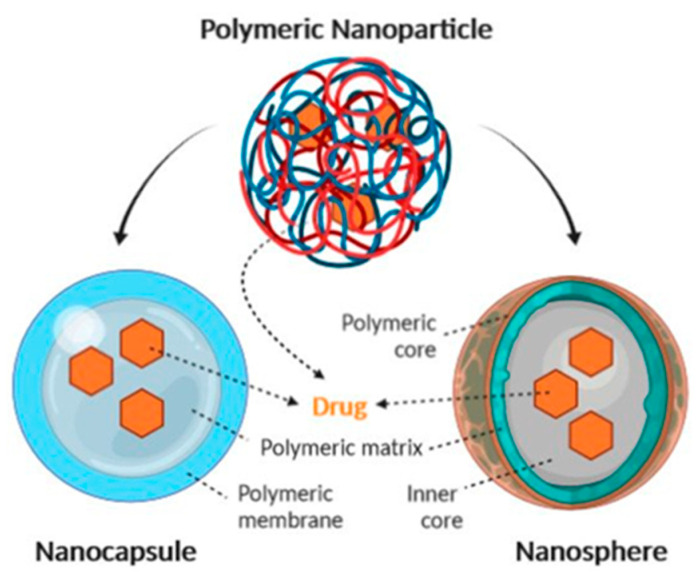
Schematic representation of nanocapsule and nanosphere structure [71].

**Table 1 ijms-22-06538-t001:** Selected TKIs, their molecular targets, FDA approval years, number of clinical trials and treated diseases.

Compound	Target	Number of Clinical Trials	Diseases	FDA Approval
Imatinib	Abl, PDGFR, Kit	754	CML, GIST, GVHD, many hematological and solid tumors	2001
Dasatinib	Abl, PDGFR, Kit, Src	320	CML, ALL, lymphoma, NSCLC and others solid tumors	2006
Nilotinib	Abl, PDGFR, c-Kit, LCK, EPHA3, EPHA8, DDR1, DDR2, MAPK11, ZAK	219	CML, ALL, GIST	2007
Ponatinib	Abl, Src, FGFR, PDGFR, VEGFR,	67	CML, ALL	2012
Asciminib	Abl	13	CML	//
Flumatinib	Abl, PDGFR, c-Kit, CSFR	5	CML	//
Sunitinib	PDGFR, Kit, FLT3, VEGFR, CSF1R	610	RCC, GIST	2006
Sorafenib	PDGFR, c-Kit, FLT3, VEGFR, B-Raf	870	RCC, liver and thyroid cancers	2007
Axitinib	Abl, PDGFR, VEGRF, c-Kit	161	RCC	2012
Ruxolitinib	JAK1, JAK2	258	Myelofibrosis, polycythemia vera, GVHD, many other different diseases	2011

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
