# Peer review of "Nanotechnology of Tyrosine Kinase Inhibitors in Cancer Therapy: A Perspective"

_ijms, 2021, doi:10.3390/ijms22126538_

Round 1
Reviewer 1 Report
This is a very interesting and well written review related with delivery systems for small molecules tyrosine kinase inhibitors in cancer therapy. However, some parts must be completed or cleared.
Line 154, shade EPR effect (Danhier F. To exploit the tumor microenvironment: Since the EPR effect fails in the clinic, what is the future of nanomedicine? J Control Release. 2016 Dec 28;244(Pt A):108-121. doi: 10.1016/j.jconrel.2016.11.015. Epub 2016 Nov 18. PMID: 27871992.)
Line 169 PEGylation do not lead always to an increasing hydrodynamic volume (depending on PEG chain length), please delete this part. PEG coating reduces the macrophage uptake by creating a hydration layer that prevents the non-specific adsorption of opsonins onto particles and reduces cell adhesion. It is not clearly explained, please modify the part 2 Nanoparticles for cancer treatment.
In the part3 “Nanoparticles of tyrosine inhibitors”, the reference 77 and 78 are not adequate because they are not related to “new formulations” in general but are focused on liposomes.
Author Response
I thank the reviewer for the timely and constructive comments.
Line 154, shade EPR effect (Danhier F. To exploit the tumor microenvironment: Since the EPR effect fails in the clinic, what is the future of nanomedicine? J Control Release. 2016 Dec 28;244(Pt A):108-121. doi: 10.1016/j.jconrel.2016.11.015. Epub 2016 Nov 18. PMID: 27871992.)
In line 152 the bibliographic reference and the comment on the suggested paper have been inserted
Line 169 PEGylation do not lead always to an increasing hydrodynamic volume (depending on PEG chain length), please delete this part. PEG coating reduces the macrophage uptake by creating a hydration layer that prevents the non-specific adsorption of opsonins onto particles and reduces cell adhesion. It is not clearly explained, please modify the part 2 Nanoparticles for cancer treatment.
The sentence “increasing hydrodynamic volume” has been removed and the explanation relating to pegylation has been added in line 172.
In the part3 “Nanoparticles of tyrosine inhibitors”, the reference 77 and 78 are not adequate because they are not related to “new formulations” in general but are focused on liposomes.
I apologize, there was a transcription error, I have inserted the new patent references in the bibliography [91,92] in line 404.
Reviewer 2 Report
The article entitled „Nanotechnology of tyrosine kinase inhibitors in cancer therapy: a perspective” collects and summarizes the potentially applicable nano carrier systems for tumor therapy in case of tyrosine kinase cargo. The article is well written, easy to read, however I have few comments and questions which should be clarified:
- The manuscript contains few typos (with red colour), which should be corrected e.g. line 8, 100, 148, 537.
- In line 159-162 it is mentioned nanoparticles with specific ligands enables the interaction to tumor cells. Please specify which ligands can be suitable for that purpose, or is there any trend, which type of tumor-specific moiety are commonly applied?
- Section 2.1.1 mentions the advantages of silica nanoparticles. Please also elaborate the disadvantages of that type of nanocarrier, as toxicity on the immune system or silicosis in case of pulmonary administration.
- Section 2.1.2 is too general, please specify how these drug delivery systems can be applied for targeted drug delivery? e.g. application of pH sensitive liposomes etc.
- Line 288 mentions Doxil® as firstly approved liposomal formulation. Please also add the active substance of the formulation.
- Line 290 describes the general components of a liposome. It is important to mention, the wall-forming phospholipids should contain two hydrophobic tail this is what sets it apart from a noisome.
- Line 344-345 describes albumin as suitable nano carrier. Well, this is very important to mention the advantages of albumin-based drug delivery systems, moreover additional information should be added to that section. Not to be overlooked, Abraxane® is an approved formulation applied in cancer therapy. By the way, albumin belongs rather to protein-based drug delivery systems, than polymeric-based, therefor I suggest to add an additional section for it.
- It would be very useful to add through which administration routs where these TKI formulations investigated, are there any research focusing on alternative delivery routes?
Author Response
I thank the referree for the suggestions proposed.
- The manuscript contains few typos (with red colour), which should be corrected e.g. line 8, 100, 148, 537.
I apologize there were some corrections highlighted, I made the changes.
- In line 159-162 it is mentioned nanoparticles with specific ligands enables the interaction to tumor cells. Please specify which ligands can be suitable for that purpose, or is there any trend, which type of tumor-specific moiety are commonly applied?
Examples of ligands have been included in lines 159, 161 and 162 with the relative bibliographic references [34,35,36].
- Section 2.1.1 mentions the advantages of silica nanoparticles. Please also elaborate the disadvantages of that type of nanocarrier, as toxicity on the immune system or silicosis in case of pulmonary administration.
In line 233 possible disadvantages of silica NPs have been included [51].
- Section 2.1.2 is too general, please specify how these drug delivery systems can be applied for targeted drug delivery? e.g. application of pH sensitive liposomes etc.
In line 313, examples of targeted liposomes have been inserted [63,64,65].
- Line 288 mentions Doxil® as firstly approved liposomal formulation. Please also add the active substance of the formulation.
I apologize for forgetting the description of the Doxil®, I also added the Caelix®(line 294-296 [58]).
- Line 290 describes the general components of a liposome. It is important to mention, the wall-forming phospholipids should contain two hydrophobic tail this is what sets it apart from a noisome.
In line 296, I mentioned the niosome [59] as requested.
- Line 344-345 describes albumin as suitable nano carrier. Well, this is very important to mention the advantages of albumin-based drug delivery systems, moreover additional information should be added to that section. Not to be overlooked, Abraxane® is an approved formulation applied in cancer therapy. By the way, albumin belongs rather to protein-based drug delivery systems, than polymeric-based, therefor I suggest to add an additional section for it.
In line 390, I have added a paragraph “Albumin-based nanocarriers” in which I have inserted the description of Abraxane®[88,89].
- It would be very useful to add through which administration routs where these TKI formulations investigated, are there any research focusing on alternative delivery routes?
I added a comment on the line 406 on possible alternative routes of administration [93,94].

Round 2
Reviewer 2 Report
The authors have addressed all my concerns and therefore I support publication without further changes